# Young children's footwear taxonomy: An international Delphi survey of parents, health and footwear industry professionals

Cylie M. Williams[1,2¤a]*, Stewart C. Morrison[3¤b], Kade Paterson[4¤c], Katherine Gobbi[5], Sam Burton[6], Matthew Hill[2¤d], Emma Harber[7], Helen Banwell[8¤e]

1 School of Primary and Allied Health Care, Faculty of Medicine, Nursing and Health Science, Monash University, Frankston, Victoria, Australia, 2 Centre for Biomechanics and Rehabilitation Technologies, Staffordshire University, Staffordshire, United Kingdom, 3 School of Life Course and Population Sciences, King's College, London, United Kingdom, 4 Centre for Health, Exercise and Sports Medicine, Department of Physiotherapy, School of Health Sciences, Faculty of Medicine Dentistry & Health Sciences, The University of Melbourne, Melbourne, Victoria, Australia, 5 Parent (Consumer Representative), Melbourne, Victoria, Australia, 6 Bobux International, Newmarket, Auckland, New Zealand, 7 Parent (Consumer Representative), Church Stretton, Shropshire, United Kingdom, 8 Allied Health and Human Performance, University of South Australia, Adelaide, South Australia, Australia

¤a Current address: School of Primary and Allied Health Care, Monash University, Frankston, Victoria, Australia
¤b Current address: King's College London, Strand, London, United Kingdom
¤c Current address: Centre for Health, Exercise and Sports Medicine, The University of Melbourne, Victoria, Australia
¤d Current address: Centre for Biomechanics and Rehabilitation Technologies, Staffordshire University, Stoke on Trent, United Kingdom
¤e Current address: Allied Health and Human Performance, City East Campus, University of South Australia, Adelaide, South Australia, Australia
* Cylie.williams@monash.edu

**Data Availability Statement:** All data available to be freely shared is within the paper and its Supporting Information files. Additionally,

## Abstract

### Objective

There is little consistency between commercial grade footwear brands for determining shoe sizing, and no universally accepted descriptors of common types or features of footwear. The primary aim of this research was to develop a footwear taxonomy about the agreed types of footwear commonly worn by children under the age of six. Secondary aims were to gain consensus of the common footwear features, when different types of footwear would be commonly worn, common terms for key footwear parts, and how movement at some of these footwear parts should be described.

### Materials and methods

Opinions were collected through a three-round modified Delphi international online survey from parents, health professionals, researchers, and footwear industry professionals. The first survey displayed generic pictures about different footwear types and asked participants to provide a grouping term, when the footwear would be worn (for what type of activity) and any grouping features. The second and third rounds presented consensus and gathered agreement on statements.

participant level data is now stored our the university data repository: https://doi.org/10.26180/19836160.v1

**Funding:** The authors received no specific funding for this work.

**Competing interests:** CMW, HB, KP, MH, SM, KG and EH have not received any support from any organisation for the submitted work, nor financial relationships with any organisations that might have an interest in the submitted work. SB is employed by Bobux International. Employment does not alter our adherence to PLOS ONE policies on sharing data and materials.

## Results

There were 121 participants who provided detailed feedback to open-ended questions. The final round resulted in consensus and agreement on the names of 14 different footwear types, when they are commonly worn and their common features. Participants also reached consensus and agreement on the terms *heel counter* to describe the back part of footwear and *fixtures* as the collective term for features allowing footwear adjustability and fastening. They also agreed on terms to quantify the flexibility at footwear sole (bend or twist) or the heel counter.

## Conclusion

This first taxonomy of children's footwear represents consensus amongst different stakeholders and is an important step in promoting consistency within footwear research. One shoe does not fit all purposes, and the recommendations from this work help to inform the next steps towards ensuring greater transparency and commonality with footwear recommendations.

## Introduction

The commercial grade footwear industry has emerged as a global business, with a market reach of approximately US$360,000 million (US) in 2020, and an increasing annual growth rate of over 5% per year [1]. The footwear industry is complex, with small and large companies co-existing, often purporting design differences or mechanical properties as their 'edge' within a competitive market. As such, there is little consistency between commercial grade footwear brands for determinants of shoe sizing, and no universally accepted descriptors of common types or features of footwear [2]. This can be problematic when specific footwear features are desired or prescribed by health professionals as part of a therapeutic intervention, which potentially comes into conflict with any footwear benefits promoted by a footwear company.

Children's footwear represents 18% of the commercial grade footwear sector [1] and plays an important role in protecting and supporting the growing foot [3]. This is of particular importance in the younger child, from new walkers until around 6 years of age, as they typically engage in increasingly complex bipedal activities during a time of increased tissue plasticity [4]. The purchase of children's footwear is a common source of angst for parents and caregivers [5], with ill-fitting and poor choice of footwear often cited as the basis of foot-related issues as adults [6]. This angst can be heightened when children present with disability or developmental concerns, where specific footwear features may assist in achieving, improving or maintaining ambulation [7–10]. The lack of consistency in descriptors of footwear types and features can stymie both health professionals and parents as it is typically dependant on the individual retail centre to interpret prescribed or recommended inclusions. Additionally, this lack of established descriptors limits the ability to confidently compare research outcomes when investigating the impact of footwear given type and features cannot be robustly described [2, 7].

The primary aim of this research was to develop a footwear taxonomy through international consensus about the types of footwear commonly worn by children under the age of six. Secondary aims were to gain consensus of the common footwear features, when different types of footwear would be commonly worn, common terms for key footwear parts, and how movement at some of these footwear parts should be described.

## Materials and methods

### Design

The study was an international three-round modified Delphi online survey. This design consisted of an initial round where participants' provided their opinion to gather consensus [11]. Any responses not reaching consensus were then returned to participants for consideration, and rating agreement in subsequent rounds. This research was approved the Monash University Human Research Ethics Committee (25698). All participants provided written informed consent through their response to the online survey.

### Participants

Participants were recruited through institutional and personal social media accounts of the authors. Participants were eligible to be part of the Delphi survey if they self-identified in any of the following categories:

1. A parent of a child/children under the age of six years and had purchased shoes for their child in a shoe store with fitting support.

2. A health professional who had made regular footwear recommendations for children under the age of six years, in the past six months.

3. A researcher who had researched young children's footwear in the past 10 years

4. A professional who had sold footwear in the past six months for children under the age of six years.

5. A professional who had worked in footwear design for children under the age of six years in the last six months.

Advertisements to encourage participation were customized to health professionals, researchers, parents of children under the age of six, and people working in the footwear industry directly relating to footwear for young children. These were advertised on social media at weekly intervals during Round 1.

There were no geographical boundaries to recruitment. Participants checked an online consent box for ongoing communication as part of the research, and to signify their commitment to responses to all rounds. Intra-panel communication was anonymous, and participants were asked to keep their responses in each round confidential. No enticements or compensation were provided.

### Procedure

A purpose-built survey was developed by the authorship team due to the novelty of the questions of interest. Face validity was tested during development by collecting multiple photos of footwear types from those currently in online advertising in Australia, the United Kingdom and the United States. All authors then reviewed pictures of the types of footwear and agreed on grouping styles, that all grouping styles were represented, and the question phrasing for the target participants. The authorship group consisted of five clinician researchers, two parents with no research experience and one industry representative. All authors participated in all rounds of survey designs. Round 1 survey was then piloted with one parent and two health professionals and wording clarified based on their feedback.

All data were collected using the online survey platform Qualtrics® software (Qualtrics, Provo, UT, USA). Data were linked at each round through participant-provided email.

Qualtrics® routinely collects Internet Protocol (IP) addresses as part of the de-identified metadata in the survey response and participants were provided with this information as part of their informed consent. The IPs were only viewed and used as a last resort (1 occasion) to match data where emails responses in subsequent rounds did not match those in the Round 1. All rounds were open for four calendar weeks and participants were reminded weekly.

Feedback to participants at each round was provided within the online survey and participants were invited to provide feedback on terminology or grammar. Final results were provided to all participants if they completed all rounds.

**Round 1.** Participants self-selected the group they identified with and were able to select more than one if it was applicable. Participants were asked to provide their gender and residing country. Based on the group selection, additional information was collected using survey software logic. This meant that only the questions relevant to the selected group were displayed. For example, if they identified as a parent, they were asked how many children they had, and the age of their youngest child. Health professionals were asked to provide their profession, how many children treated in a typical month who were aged under six years, and how long they had been working in the role. Researchers, footwear designers and those working in footwear retail were also asked how many years they had been working in their role.

Participants were then progressed through the first round of the online survey. The survey presented participants with three pictures of similar footwear that had similar features (S1 File). These footwear pictures and their groupings were co-designed by all authors based on their expertise as consumers, health professionals or footwear designers. No brands were shown, and all footwear pictures were of shoe styles available in the countries of the authors. An example of the figures is displayed in Fig 1.

For each footwear group picture, participants were asked the following questions (and prompt for the question was placed in *italics*).

1. When you look at the pictures, what would you call this group of footwear?
   (This may be a simple response and we'd urge you to consider the first grouping word that comes to mind.)

2. When do children usually wear this type of footwear?
   (This may be related to a particular time of year, a season or seasonal activity or when a child does a particular activity where they would commonly wear this type of footwear for.)

3. What are the common features of this group of footwear?
   (We'd encourage you to be as detailed as possible, and list as many features are you can think of. Features are like how high the shoes are, what the top or bottom of the footwear looks like, as well as the front and back of the footwear.)

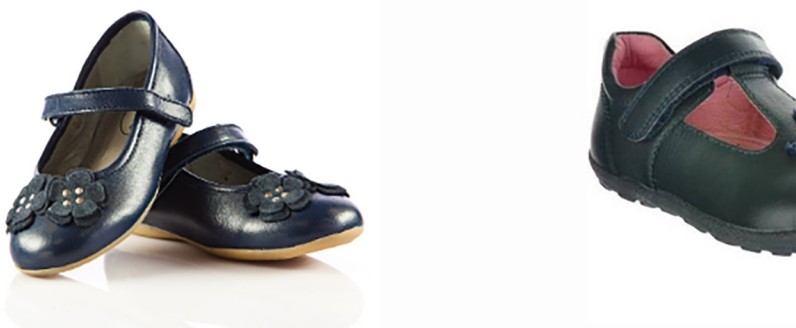

**Fig 1. Footwear style example.**

4. What are some of the other names you have heard these features called, with similar features to these?
   (This may be what the store calls them, what your parents, friends or interstate or international colleagues call them.)

Questions within the survey were specifically designed to not prompt any terms or infer responses for future questions.

Participants were then invited to describe any other footwear types young children commonly wear that were not displayed in the pictures. Participants were shown three pictures of footwear with different responses to torsion applied to the sole of the footwear, three pictures of different responses to pressure applied at the back of the heel of the footwear and a picture of different footwear with adjustable features. Participants were asked to describe a group term for these features shown in the pictures. An example of these is shown in Fig 2.

To develop Round 2, participant responses were initially grouped into a) Health professionals and researchers, b) Parents and c) Footwear industry professionals, based on the numbers of responses. Where participants selected more than one category, they were allocated to the category on the hierarchal order based on training and exposure to footwear and where health professionals and researchers were set at the highest category. For example, if a participant responded that they were a health professional, parent and sold footwear, they were placed in the health professional grouping.

Inductive quantitative content analysis of the open questions was undertaken. This method of analysis allowed for statements and comments to be individually considered, the content of these statements based on what is commonly understood about footwear and a statement made with common themes [12]. This approach meant that the first participant's comment was considered, and one or more statements developed from this. The next comment was then considered and counted towards that statement or a new statement generated. As anticipated, the length of statements varied, however, even where the statement was one word, it was counted to a statement or a new statement generated. This grouping took an iterative approach, whereby if a new statement emerged, earlier comments were recoded.

The data were initially analysed by a single researcher (CW). To reduce individual bias, the participant comments and statements were independently reviewed by an additional author, with all other authors reviewing at least 5 comments each. Each author provided secondary review based on their knowledge and own personal grouping (Health professionals–HB, SM, KP, MH, parents–KG, EH or footwear industry—SB). Disagreements were resolved by discussion. Reflexivity was acknowledged as a concept that introduces personal bias into research

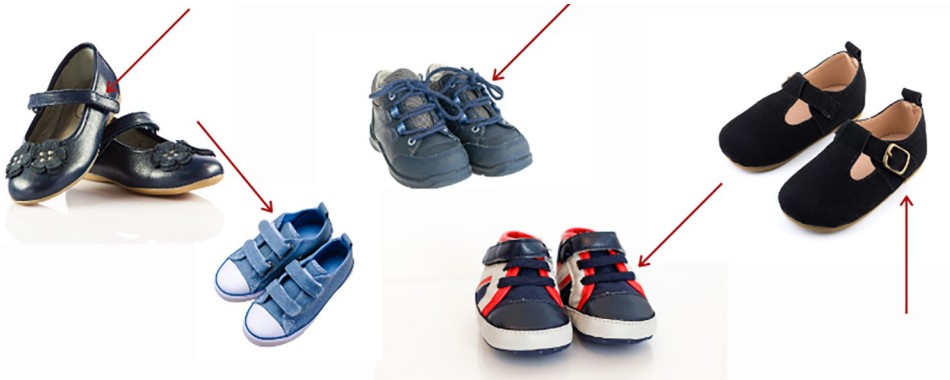

**Fig 2. Footwear features example.**

[13]. Authors analysing this data acknowledged their different individual experiences with children's footwear, purchasing, knowledge, and how these different experiences may have influenced the analysis.

**Round 2.** Statements presented to participants in Round 2 (S2 File) were considered to have reached consensus within Round one when 70% or more of the participants in each group indicated the same statement content by agreement of two authors. This percentage was consistent with existing literature [14, 15]. Participant groupings were used to ensure equal consideration of the views of all participants for Round 1 only. This subgrouping was used to ensure one grouping did not unduly influence the results based on participant numbers.

Only statements arising from 50–69% of participants in total or within any subgroup were presented to participants in Round 2 for agreement rating. Participants were made aware when there was less than 50% of the total number of responses, but where there was a group that had a 50% or greater response. They were not informed which group reached 50% or greater so as not to influence any bias or value judgement placed on the statement. Statements where less than 50% of participants in any group responded similarly were discarded and did not appear in Round 2.

Participants were then asked to indicate their agreement with each statement on a four point Likert scale where 1 was Strongly Disagree, 2 was Disagree, 3 was Agree and 4 was Strongly Agree. They were also asked to provide suggestions to grammar or statement wording if they did not agree with the statement.

**Round 3.** Similar to the process in Round 2, statements where 70% or more participants agreed or strongly agreed were included (S3 File). It was planned that statements where less than 70% of participants agreed were discarded from the final result, consistent with the Delphi survey process.

## Analysis

Descriptive statistics and analysis of responses of each round were undertaken in Microsoft Excel 2018 (Microsoft Corp, Redmond Washington). The authors made *a priori* decision that the Delphi would conclude if the total or sub-group participant response rate dropped below 70%, or if round three was required and completed, irrespective of agreement. Participants who did not complete the entire questions in Round 1 were excluded and not invited to complete Rounds 2 and 3.

## Results

### Participants

There were 159 participants who consented to complete the first round of the Delphi survey. Of these, there were 121 completed responses. Demographics of included participants and their sub-groupings are provided in Table 1. Table 2 provides further details about the health professionals who participated, including a breakdown of the professions, average number of young children treated per month and years of experience. The number of participants in each round is shown in Fig 3. There were 55 (45% of 121) participants who had children <6 years of age, and nine participants who worked in the footwear industry, four of these were also health professionals. The median (IQR) number of children was 2 (1, 2) and the median child age was 3 (1,4) years.

### Consensus

Round 1 took participants approximately 60 minutes to complete. Participants generated 147 statements in response to open ended questions. There were 16 consensus statements about the names for footwear styles, when they are worn, their common footwear features. Tables 3

**Table 1. Demographics of participants and allocations to subgroups.**

|  | Total participants (N = 121) |
|---|---|
| Country, n(%) |  |
| Australia | 65 (54%) |
| United Kingdom | 30 (25%) |
| USA | 11 (9%) |
| Other* | 15 (12%) |
| Female, n(%) | 98 (91%) |
| Health professionals, n (%) | 90 (74%) |
| Researchers, n (%) | 1 (1%) |
| Working in the footwear industry | 9 (7%) |
| Parents of children <6 years of age | 55 (45%) |
| **Participant Grouping 1** *(Health professionals and researchers who may also sell, design footwear or also be parents)* n (%) | 90 (74%) |
| **Participant Grouping 2** *(Parents only)* n (%) | 26 (21%) |
| **Participant Grouping 3** *(People who sell and design footwear only)*, n (%) | 5 (5%) |

*Canada, Malta, Singapore, Denmark, New Zealand

and 4 provide all statements generated by participants and the frequency (%) of participants who provided the same response. Statements highlighted in green were those that meet consensus (≥70% of all participants providing the same response). There were 71 statements where less than 50% of participants described content, these were discarded at Round 1, and highlighted in red in Tables 3 and 4. Statements highlighted in orange in Tables 3 and 4 were developed from similar statements from 50–69% of participants in total, or in each participant group and progressed to the next round.

## Agreement

Round 2 took approximately 25 minutes and Round 3 less than 5 minutes for participants to complete. Tables 3 and 4 provide an outline of the statements progressing through Round 2 and Round 3 using the frequency (%) and same colour coding system as Round 1. In Round 2,

**Table 2. Health professional participant demographics (n = 90).**

|  | Total health professionals (n = 90) |
|---|---|
| Podiatrist | 42 (47%) |
| Physiotherapist or Physical Therapist | 40 (44% |
| Orthotist or Pedorthist | 8 (9%) |
| Number of children treated in a typical month |  |
| <10 | 29 (33%) |
| 10–19 | 24 (27%) |
| >20 | 37 (40%) |
| Years of experience |  |
| <5 years | 10 (11%) |
| 5–9 years | 21 (23%) |
| 10 or more years | 59 (66%) |

**Fig 3. Participant flow through rounds.**

there were 57 statements with 70% or greater agreement, one statement with 50–59% agreement and one statement discarded. The final statement reached agreement in Round 3. On review of the consensus results, it was acknowledged that participants were presented with one statement in Rounds 2 and 3 that should have been discarded following Round 1, and this been acknowledged in Table 2.

**Table 3. Generated statements about young children's footwear taxonomy and round in which the statements were accepted.**

| Domain | Statement | Round 1 n (%) of 121 participant responses* n (%)[a-90, b-26 or c-5] | Round 2 (n) % of 105 participant responses | Round 3 n (%) of 102 participant responses |
|---|---|---|---|---|
| Style | **Boot** | 114 (94) | | |
| Worn | Boots are often worn when it is cold, wet, snowing, or in winter. | 115 (95) | | |
| Worn | Boots are often worn when going outdoors | 65 (54) | 96 (91) | |
| Worn | Boots are often worn during physical activity such as walking, hiking or climbing | 48 (40) 3 (60)[c] | 96 (91) | |
| Worn | Boots are prescribed by health professional for foot support or a foot problem | 30 (25) | | |
| Feature | Boots cover the ankle | 106 (88) | | |
| Feature | Boot sole has a tread pattern with a variable heel height and/or width | 37 (31) | | |
| Feature | The boot sole is commonly made of a material that resists bending | 69 (57) | 84 (79) | |
| Feature | The boot upper material covers the toes and foot | 53 (44) 45 (50)[a] | 100 (94) | |
| Feature | The boot upper material is commonly leather or a material that can be waterproofed | 53 (44) 4 (80)[c] | 91 (86) | |
| Feature | Boots commonly have fastenings or elastic to improve their fit | 53 (44) 45 (50)[a] | 86 (81) | |
| Feature | Boots are structured around the heel | 38 (31) | | |
| Style | **Sneaker** | 62 (51) | 75 (71) | |
| Style | Plimsol | 19 (16) | | |
| Style | Joggers | 5 (4) | | |
| Style | Runners | 31 (26) | | |
| Worn | Sneakers are commonly worn when being physically active, or for casual occasions. These activities or occasions may include play, or event-based occasions (e.g. family gatherings). | 85 (70) | | |
| Worn | Recommended by a health professional for its benefit or a foot problem | 2 (2) | | |
| Worn | Sneakers are commonly worn when the weather is dry or warm | 62 (51) | 87 (82) | |
| Worn | Sneakers are commonly worn outdoors | 75 (62) | 93 (88) | |
| Worn | Worn at a particular age or stage | 28 (23) | | |
| Feature | Sneakers commonly have a soft or very flexible sole | 87 (72) | | |
| Feature | Sneakers commonly have an upper material fully covers the top of the foot | 86 (71) | | |
| Feature | Sneakers commonly have a heel counter that has some structure and stiffness | 76 (63) | 75 (71) | |
| Feature | Sneakers commonly have fasteners such as Velcro or laces to adjust the fit to the foot | 69 (57) | 99 (93) | |
| Feature | Upper is commonly canvas or leather | 57 (47) | | |
| Feature | Light weight | 15 (12) | | |
| Feature | Footwear finished under the ankle | 41 (34) | | |
| Feature | Limited or no arch support | 2 (2) | | |
| Feature | Light weight | 15 (12) | | |
| Style | **Sneaker** | 85 (70) | | |
| Style | **Runner** | 78 (64) | 81 (76) | |
| Style | **Sport/athletic footwear** | 42 (35) 3 (60)[c] | 98 (92) | |
| Style | Jogger | 32 (26)[β] | | |
| Style | Trainers | 49 (40) | | |
| Style | Sand shoes | 8 (7) | | |
| Style | Pumps/takkies or kicks | 3 (2) | | |
| Style | Brand or sport specific shoe name | 20 (17) | | |

(Continued)

**Table 3.** (Continued)

| Domain | Statement | Round 1 n (%) of 121 participant responses* n (%)[a-90, b-26 or c-5] | Round 2 (n) % of 105 participant responses | Round 3 n (%) of 102 participant responses |
|---|---|---|---|---|
| Worn | This style of footwear is commonly worn when being very active, such as running or playing sport | 116 (96) | | |
| Worn | This style footwear is commonly worn in all seasons | 43 (36) 3 (60)[c] | 99 (93) | |
| Worn | This style of footwear is commonly worn outdoors or during organised care (e.g. nursery school or kindergarten) | 67 (55) | 100 (94) | |
| Worn | Worn on recommendation by a health professional for its benefit or features | 5 (4) | | |
| Worn | This style of footwear can be worn everyday | 40 (33) 3 (60)[c] | 98 (92) | |
| Worn | Worn at a particular age or developmental stage | 3 (2) | | |
| Feature | This style of footwear has semi-flexible sole made of cushioned material | 82 (68) | 88 (83) | |
| Feature | The bottom or sole of this type of footwear has a gripping tread, and is higher underneath the bottom of the heel area than underneath the front area | 57 (47) 46 (51) | 90 (85) | |
| Feature | The upper material of this style of footwear covers the top of the foot | 64 (53) | 101 (95) | |
| Feature | This style of footwear has fasteners to adjust fit | 79 (65) | 99 (93) | |
| Feature | Footwear finishes under the ankle | 35 (30) | | |
| Feature | The footwear is light weight | 10 (8) | | |
| Feature | The material of the upper has features to improve breathability/airflow | 44 (36) | | |
| Feature | This style of footwear commonly has a structured and semi-flexible heel counter | 77 (64) | 99 (93) | |
| Feature | The footwear has an insole that is moulded or has an arch contour | 23 (19) | | |
| Style | **Sandal** | 100 (83) | | |
| Style | Summer shoes | 13 (11) | | |
| Style | Miscellaneous terms such as Slip ons, jandals, jellies, beach shoes, open toe shoes, thongs, slides | 25 (21) | | |
| Worn | Sandals are commonly worn during summer, or in warm weather | 98 (81) | | |
| Worn | Sandals are commonly worn outside to places like the beach, or for casual outings | 74 (61) | 97 (92) | |
| Worn | Worn for particular play activities such as wet play or outdoor play | 24 (20) | | |
| Worn | Worn at particular ages or for every day | 3 (2) | | |
| Feature | Sandals commonly have upper material that has gaps or holes, and the material may or may not totally cover the toes | 97 (80) | | |
| Feature | Sandals commonly have a semi flexible flat sole | 56 (46) 3 (60)[c] | 88 (83) | |
| Feature | Sandals can have either a strap at the heel or an enclosed back | 69 (57) | 100 (94) | |
| Feature | The upper material of sandals is commonly either leather or synthetic material | 23 (19) 3 (60)[c] | 101 (95) | |
| Feature | Sandals commonly have a strap around the front of the ankle that can be adjusted for fit | 78 (64) | 95 (90) | |
| Feature | Footwear finishes under the ankle | 18 (15) | | |
| Feature | The footwear is light weight | 7 (6) | | |
| Style | **Pre-walkers** | 52 (43) 4 (80)[c] | 94 (89) | |
| Style | Booties | 20 (22) | | |
| Style | **Soft-soled footwear** | 19 (16) 5 (100)[c] | 96 (91) | |
| Style | Moccasins | 16 (13) | | |
| Style | Baby shoes | 50 (41) | | |
| Style | Slippers | 28 (23) | | |

*(Continued)*

**Table 3.** (Continued)

| Domain | Statement | Round 1 n (%) of 121 participant responses* n (%)^a-90, b-26 or c-5 | Round 2 (n) % of 105 participant responses | Round 3 n (%) of 102 participant responses |
|---|---|---|---|---|
| Worn | Pre-walkers or soft-soled footwear can be worn by babies or children not yet confidently walking | 73 (60) | 98 (92) | |
| Worn | Footwear can be worn as a fashion accessory or item | 12 (10) | | |
| Worn | Pre-walkers or soft-soled footwear can be worn indoors or during organised care (i.e. Nursery or daycare) | 26 (21) 3 (60)^c | 92 (87) | |
| Worn | Pre-walkers or soft-soled footwear can be worn while learning a new skill such as crawling or walking | 48 (40) 3 (60)^c | 83 (78) | |
| Worn | Pre-walkers or soft-soled footwear can protect feet from the environment or the cold | 37 (31) 3 (60)^c | 98 (92) | |
| Feature | Pre-walker or soft-soled footwear has a soft and fully flexible sole | 111 (92) | | |
| Feature | The upper material and heel area (heel counter) of pre-walkers or soft-soled footwear are fully flexible | 101 (83) | | |
| Feature | The upper of pre-walkers or soft-soled footwear is either made of leather, fabric or a synthetic material that is soft. | 3 (26) 3 (60)^c | 101 (95) | |
| Feature | Footwear finishes under the ankle | 8 (7) | | |
| Feature | Footwear has some form of fixation to keep the shoe on the foot | 18 (15) | | |
| Feature | Footwear commonly slips onto the foot | 18 (15) | | |
| Style | **Mary-Jane** | 80 (66) | 86 (91) | |
| Style | **T-Bar** | 25 (21) 3 (60)^c | 68 (64) | 71 (70) |
| Style | Dress shoe | 30 (25) | | |
| Style | Court. Formal, dolly or party shoe | 13 (11) | | |
| Style | Ballet flats or pumps | 42 (35) 4 (80)^c | 46 (43) | |
| Style | School or church shoes | 35 (29) | | |
| Worn | This type of footwear is commonly worn indoors, or during organised care (i.e. Nursery school or kindergarten) | 80 (66) | 85 (80) | |
| Worn | This footwear is commonly worn during special, or more dressy occasions | 83 (69) | 95 (90) | |
| Worn | This type of footwear can be worn in variable temperatures and seasons | 25 (21) | | |
| Worn | This type of footwear is commonly worn when not being physically active | 12 (10) | | |
| Feature | This footwear covers the toes, but does not fully cover the top of the foot, and is secured by a strap at the ankle | 93 (77) | | |
| Feature | This footwear commonly has a flat and non-slip sole | 51 (42) 13 (50)^b | 86 (81) | |
| Feature | This footwear has a thin sole with variable flexibility to bending | 46 (38) | | |
| Feature | The upper material of the footwear is commonly either made of leather or synthetic materials, which has a rounded shape over the toes | 41 (34) 3 (60)^c | 98 (92) | |
| Feature | The footwear finishes under the ankle | 31 (26) | | |
| Feature | The footwear is light weight | 4 (3) | | |
| Feature | The footwear has no arch support | 5 (4) | | |
| Style | **Boat shoes** | 57 (47) 14 (54)^b | 93 (88) | |
| Style | Slip-ons | 36 (30) | | |
| Style | **Loafers** | 67 (55) | 99 (93) | |
| Style | Moccasins | 28 (23) | | |
| Worn | Worn during warmer weather | 39 (32) | | |
| Worn | Boat shoes or loafers are commonly worn during a special or more formal occasion | 85 (70) | | |

*(Continued)*

**Table 3.** (Continued)

| Domain | Statement | Round 1<br>n (%) of 121<br>participant responses[*]<br>n (%)[a-90, b-26 or c-5] | Round 2<br>(n) % of 105<br>participant responses | Round 3<br>n (%) of 102<br>participant responses |
|---|---|---|---|---|
| Worn | Worn to specific places such as school or church | 23 (19) | | |
| Worn | Worn during periods of low physical activity | 16 (13) | | |
| Worn | The footwear is for everyday use | 4 (3) | | |
| Feature | The uppers of boat shoes or loafers are commonly made of either firm leather or fabric | 36 (30)<br>4 (80)[c] | 100 (94) | |
| Feature | Footwear has a flexible flat sole | 51 (42) | | |
| Feature | The footwear upper is commonly soft | 42 (35) | | |
| Feature | The footwear covers the heel with minimal heel counter stiffness | 37 (31) | | |
| Feature | Boat shoes or loafers are commonly slip on | 74 (61) | 106 (100) | |
| Feature | The footwear cuts low under the ankles | 32 (26) | | |
| Feature | The footwear is light weight | 2 (2) | | |
| Feature | The footwear has variable fixtures | 13 (11) | | |
| Style | Thongs, flip flops, slides or jandals | 69 (57) | 98 (92) | |
| Worn | Thongs, Flip flops, slides or jandals may be worn in hot weather | ND | 100 (94) | |
| Feature | Thongs, Flip flops, slides or jandals commonly have a flexible sole and are held onto the top of the foot with a strap across the front of the foot only | ND | 93 (88) | |
| Style | Gumboots or Wellingtons | 42 (35)<br>13 (50)[b] | 100 (94) | |
| Worn | Gumboots or Wellingtons are worn in wet weather | ND | 98 (92) | |
| Feature | Gumboots or Wellingtons are made of a waterproof material | ND | 97 (92) | |
| Feature | Gumboots or Wellingtons can easily slip on and off the feet because of their shape and no fasteners | ND | 94 (89) | |
| Style | Slippers | 22 (18) | | |
| Style | Crocs | 35 (29) | | |

[*]Where total responses were <50%, but one or more groups had a 50% or greater response, the additional highest subgroup response and percentage is also provided where a) Health professionals (n = 90), b) Parents (n = 26) and c) Footwear industry (n = 5)

ND–Not displayed–worn and feature questions not displayed and the footwear styles were generated from "other" questions by participants.

Colour Legend: Red Not progress to next round, Orange Progressed to next round, Green Accepted

In Round 1, participants provided additional names of other footwear styles. Where there were regional differences in names, these were grouped. For example, flip-flops (United Kingdom and USA), Jandals (New Zealand) and Thongs (Australia) were all considered the same type of shoe. Where consensus was reached on a shoe style or group, the authors generated five new statements of their features and when they would be worn, for presentation to participants in Round 2. In developing these we reviewed the literature and reviewed pictures of the different styles.

The final footwear styles/groupings, when they were commonly worn and the features common to these styles are illustrated in a summary infographic (Figs 4 and 5) using some of the pictures throughout the survey. This infographic also provides details on preferred naming conventions of some parts of the footwear (e.g. heel counter and fasteners) and how participants described the flexibility at different parts of the footwear. For example, a picture showing a heel counter bending towards the sole >45° agreed that the amount of movement should be described as flexible with additional words to convey flexibility to a great extent such as "fully flexible" or "very flexible".

**Table 4. Generated statements about how to describe the common features of footwear and round in which the statements were accepted.**

| Domain | Statement | Round 1 n (%) of 121 participant responses* $n (\%)^{a-90, b-26 \text{ or } c-5}$ | Round 2 (n) % of 105 participant repsonses |
|---|---|---|---|
| Sole flexibility | Flexibility statement 1 when sole is able to twist >45° and bend >45° at the forefoot: The sole should be described as flexible with additional words to convey flexibility to a great extent such as "fully flexible", "extremely flexible" or "very flexible". | 86 (71) | |
| Sole flexibility | Flexibility statement 2 when sole is able to twist 10-45° and bend 10-45° at the forefoot: The sole should be described as flexible with additional words to convey flexibility to a medium extent such as "moderately flexible", "semi-flexible". | 94 (78) | |
| Sole flexibility | Flexibility statement 3 when sole is able to twist <10° and bend <10° at the forefoot: The sole should be described as flexible with additional words to convey amount such as not flexible or non-flexible. | 76 (63) | 94 (89) |
| Sole flexibility | Flexibility statement 3 when sole is able to twist <10° and bend <10° at the forefoot: The sole should be described in similar terms to convey its hardness such as: Rigid, Stiff or Solid. | 42 (35) 4 (80)^c | 92 (87) |
| Shoe back | Heel counter | 67 (55) | 96 (91) |
| Shoe back | Back of shoe | 19 (16) | |
| Shoe back | Heel cup/heel | 28 (26) | |
| Shoe back flexibility | Flexibility statement 1: Picture showing a heel counter bending towards the sole >45°: The amount of movement should be described as flexible with additional words to convey flexibility to a great extent such as "fully flexible" or "very flexible". | 55 (45) 3 (60)^c | 97 (92) |
| Shoe back flexibility | Flexibility statement 1: When heel counter bending towards the sole >45°: Soft | 27 (22) | |
| Shoe back flexibility | Flexibility statement 1: When heel counter bending towards the sole >45°: Flexible | 24 (20) | |
| Shoe back flexibility | Flexibility statement 2: Picture showing a heel counter bending towards the sole 10-45° The amount of movement should be described as flexible with additional words to convey flexibility to a great extent such as "semi-flexible" or "moderately flexible". | 67 (55) | 97 (92) |
| Shoe back flexibility | Flexibility statement 2: Picture showing a heel counter bending towards the sole 10-45° The amount of movement should be described as firm with additional words to convey firmness to a great extent such as "semi-firm" or "moderately firm". | 19 (16) | |
| Shoe back flexibility | Flexibility statement 3: When heel counter is able to bend towards the sole <10° The amount of movement should be described in similar terms to convey its hardness such as: Rigid, Stiff or Solid. | 79 (65) | 94 (89) |
| Shoe back flexibility | Flexibility statement 3: When heel counter is able to bend towards the sole <10° The amount of movement should be described in similar terms to convey its limited flexible such as non-flexible or inflexible | 23 (19) 3 (60)^c | 93 (88) |
| Adjustability collective | Fasteners | 75 (62) | 96 (91) |
| Adjustability collective | Laces | 37 (31) | |
| Adjustability collective | Straps | 28 (23) | |
| Adjustability collective | Velcro | 30 (25) | |
| Adjustability collective | Buckle | 31 (26) | |
| Adjustability collective | Closures/Fixtures | 17 (14) | |

*Where total responses were <50%, but one or more groups had a 50% or greater response, the additional highest subgroup response and percentage is also provided where a) Health professionals (n = 90), b) Parents (n = 26) and c) Footwear industry (n = 5)

Colour Legend: Red Not progress to next round, Orange Progressed to next round, Green Accepted

International consensus on types and definitions for

# Young children's footwear

by parents, health care professionals & the footwear industry

**Boots**
- worn when it is cold, wet, snowing, or in winter.
- worn when going outdoors
- worn during physical activity such as walking, hiking or climbing
- cover the ankle
- made of a material that resists bending
- upper material covers the toes & foot
- upper material is commonly leather or a material that can be waterproofed
- fastenings or elastic to improve their fit

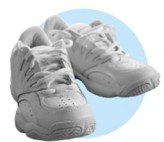

**Sneakers**
- worn when being physically active, or for casual occasions.
- worn when the weather is dry or warm
- worn outdoors
- soft or very flexible sole
- upper material that fully covers the top of the foot
- heel counter that has some structure & stiffness
- fasteners such as Velcro or laces to adjust the fit to the foot

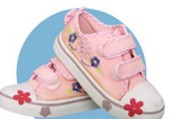

**Sneakers, Runners or Sport/Athletic footwear**
- worn when being very active
- worn in all seasons
- worn outdoors or during organised care
- worn every day
- has a semi-flexible sole made of cushioned material
- sole has a gripping tread, & is higher underneath the bottom of the heel area, than underneath the front area
- upper material covers the top of the foot
- fasteners to adjust fit
- structured & semi-flexible heel counter

**Sandals**
- worn during summer, or in warm weather
- worn outside to places like the beach, or for casual outings
- has upper material that has gaps or holes, & the material may or may not totally cover the toes
- has a semi-flexible flat sole
- have either a strap at the heel or an enclosed back
- upper material is commonly either leather or synthetic material
- has a strap around the front of the ankle that can be adjusted for fit

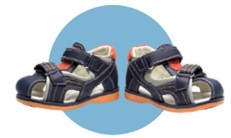

**Pre-walkers or soft soled footwear**
- worn by babies or children not yet confidently walking
- worn indoors or during organised care (i.e. Nursery or daycare)
- worn while learning a new skill such as crawling or walking
- can protect feet from the environment or the cold
- has a soft & fully flexible sole
- upper material & heel counter is fully flexible
- made of leather, fabric or a synthetic material that is soft

**Mary-Janes & T-Bars**
- worn indoors, or during organised care
- worn during special or more dressy occasions
- covers the toes, but does not fully cover the top of the foot, & is secured by a strap at the ankle
- a flat and non-slip sole
- upper material is either made of leather or synthetic materials, which has a rounded shape over the toes

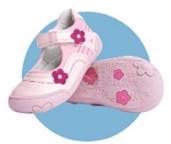

**Loafers or Boat Shoes**
- worn during a special or more formal occasion
- uppers are commonly made of either firm leather or fabric
- slip on

**Gumboots or Wellingtons**
- worn in wet weather
- made of a waterproof material
- can easily slip on and off the feet because of their shape & no fasteners

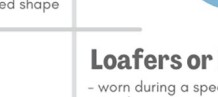

**Thongs, Flip-flops, slides or Jandals**
- worn in hot weather
- flexible sole & are held onto the top of the foot with a strap across the front of the foot only

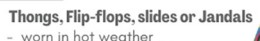
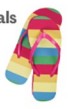

Page 1

**Fig 4. Taxonomy and common features infographic.**

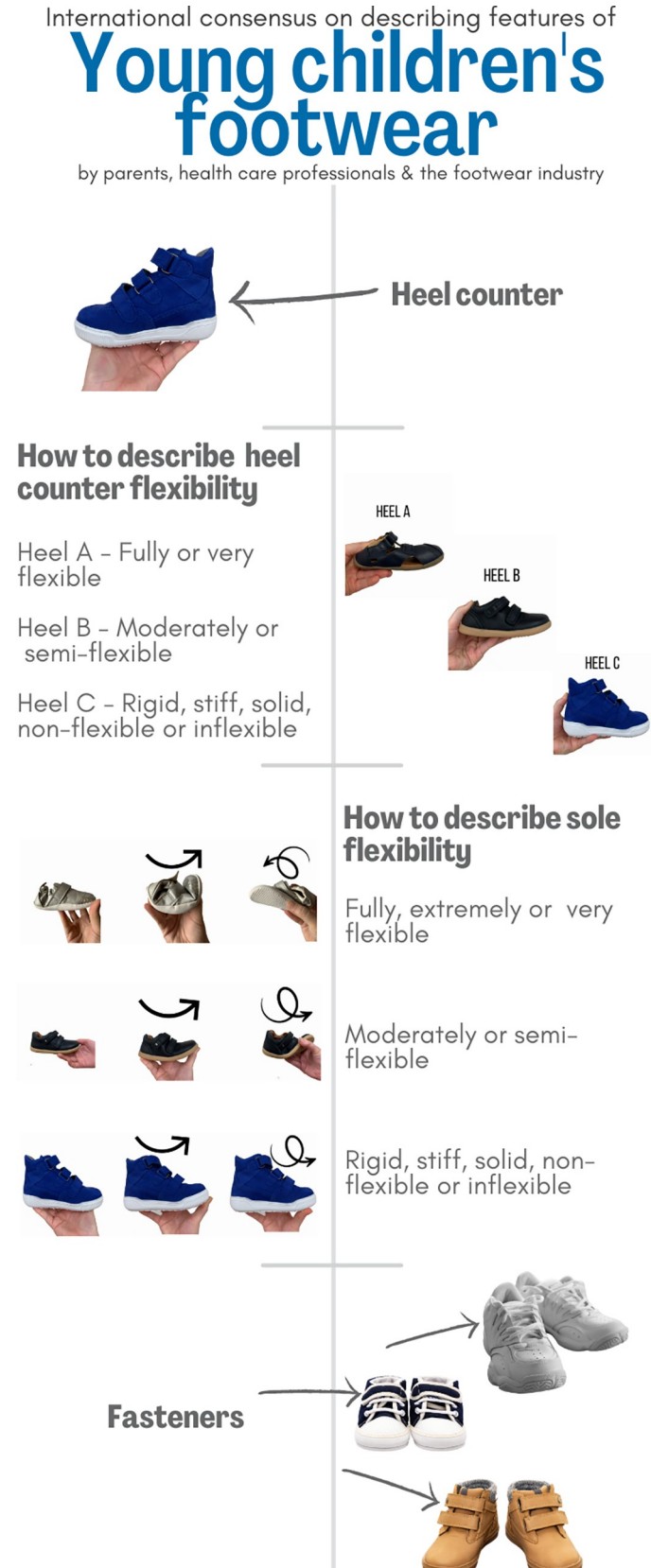

**Fig 5. Footwear feature definitions infographic.**

## Discussion

This study offers the first taxonomy for young children's footwear developed by consensus in consultation with footwear industry professionals, health professionals, and parents. This work was undertaken to respond to persistent challenges with promoting clarity about footwear information, and transparency with footwear research. Emerging from this work, Tables 3 and 4 provide consensus and agreement on footwear styles, such as what features are common in footwear called a boot. Also, when certain types of footwear are commonly worn such as when a boot is worn, and the features common to these types of footwear such as a boot commonly covers the ankle. We have also captured consensus on preferred naming conventions for components of footwear (e.g. heel counter and fasteners) and how the flexibility at different parts of the footwear are described. This taxonomy is a useful resource of contemporary terms and features of footwear, to guide terminology, research and descriptors provided in clinical practice and footwear retail.

Footwear has long been considered a factor impacting on foot development [2, 3] alongside the attainment and improvement of motor skills [16]. Whilst ongoing perspectives about footwear and association with longer-term complications is somewhat controversial, and often omits consideration of the more complex socioeconomic and cultural influences on development [17, 18], it highlights the growing interest in exploring the purpose and function of footwear in childhood and the importance of challenging long-held beliefs. Footwear is an external factor that can influence children's gait [4, 19], and differences in motor skill [20] meaning that greater consideration of footwear recommendations for toddlers and young children are required. In recent years, the focus on footwear dimensions and fit has been explored [21, 22] but there has also been a shift towards understanding the effects of footwear characteristics on biomechanical outcomes and identifying what features should typify shoes for infants and young children [4, 7]. It is acknowledged that footwear construction is multifactorial and other structural characteristics could influence foot function [8]. The consensus methods used to develop this taxonomy will underpin greater consistency with footwear description and characteristics ensuring clarity of information dissemination in future research, clinical consultations and in marketing. It will enable future researchers to describe a shoe by a term and its common features allowing reproducibility in future footwear research. Further advances in footwear research are needed to offer common understanding of terms, definitions and description of footwear to ensure that research is reproducible and supports the translation of research findings into credible recommendations.

Parents often report concerns about footwear choices for their children [5] and health professionals have an important role providing footwear education and helping parents to navigate information. We believe that the findings from this study are a prerequisite to conversations in practice about footwear choices for children, and it is hoped these findings will assist clinicians with evolving and implementing age-appropriate footwear advice, and helping parents to navigate footwear recommendations. A taxonomy will help health professionals provide accurate descriptions that are acceptable and understood by parents when prescribing footwear with an agreed statement description such as "non-flexible heel counter" or consensus statement "footwear with a moderately flexible sole". Both features are thought to provide additional stability during developing motor skills [2].

It is important to acknowledge several limitations within this study. The taxonomy was based on consensus opinion, and as manufacturing and patents often are embedded within the design of footwear, terms may differ across countries and footwear sizes. Expert consensus in the context of evidence-based practice constitutes low level evidence. We have attempted to minimise any author bias during this research by co-design with industry, health professionals,

and parents in the research team. In acknowledging the Delphi process there is also the limitation with finality and there is no guarantee of correctness [23]. Bias and "group think" of participants was minimised through confidentiality and anonymity during the process. We acknowledge limited participation from Europe and Asia. Engagement in these countries may have influenced the results and researchers are urged to consider how to ensure greater international engagement when undertaking footwear research. Lastly, we did not collect socioeconomic information from parent participants which may play a role in choices about footwear types and the opinions on how these choices impact the child. These factors could have been explored through collection of household income, education level of parent completing the survey and total number of children within the family. Withstanding this, the large number of participants and their diversity played a large role in minimising the impact of localised terms and regional footwear differences. The aim of this study was to develop a taxonomy specific to young children, and as such, generalisability may not be transferable to footwear for older children or adults.

## Conclusion

This taxonomy represents consensus amongst parents, health professionals and footwear designers and retailers, and is an important step in enabling consistency in footwear research. One shoe does not fit all purposes, and the recommendations from this work help to inform the next steps towards ensuring greater transparency and commonality with footwear recommendations. Given the enormity and complexity of the footwear industry, this study underpins the need for further work to explore footwear characteristics and further pursue clear recommendations for parents and shift away from generic recommendations with little validity.

## Supporting information

**S1 File. Round 1 survey.**
(DOCX)

**S2 File. Round 2 survey.**
(PDF)

**S3 File. Round 3 survey.**
(PDF)

## Author Contributions

**Conceptualization:** Cylie M. Williams, Stewart C. Morrison, Kade Paterson, Matthew Hill, Helen Banwell.

**Data curation:** Cylie M. Williams.

**Formal analysis:** Cylie M. Williams, Stewart C. Morrison, Kade Paterson, Katherine Gobbi, Matthew Hill, Emma Harber, Helen Banwell.

**Investigation:** Cylie M. Williams, Stewart C. Morrison, Kade Paterson, Katherine Gobbi, Sam Burton, Matthew Hill, Emma Harber, Helen Banwell.

**Methodology:** Cylie M. Williams, Stewart C. Morrison, Kade Paterson, Katherine Gobbi, Sam Burton, Matthew Hill, Emma Harber, Helen Banwell.

**Project administration:** Cylie M. Williams.

**Software:** Cylie M. Williams.

**Validation:** Stewart C. Morrison, Kade Paterson, Katherine Gobbi, Sam Burton.

**Visualization:** Emma Harber.

**Writing – original draft:** Cylie M. Williams, Stewart C. Morrison, Helen Banwell.

**Writing – review & editing:** Kade Paterson, Katherine Gobbi, Sam Burton, Matthew Hill, Emma Harber.

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
