## [Decision Letter · Decision Letter 0]

11 Aug 2021

PONE-D-21-06208

Young children’s footwear taxonomy: an international Delphi survey of parents, health and footwear professionals

PLOS ONE

Dear Dr. Williams,

Thank you for submitting your manuscript to PLOS ONE. After careful consideration, we feel that it has merit but does not fully meet PLOS ONE’s publication criteria as it currently stands. Therefore, we invite you to submit a revised version of the manuscript that addresses the points raised during the review process.

We look forward to receiving your revised manuscript.

Kind regards,

Or Kan Soh

Academic Editor

PLOS ONE

Journal Requirements:

2. Please provide additional details regarding participant consent. In the ethics statement in the Methods and online submission information, please ensure that you have specified what type of consent you obtained (for instance, written or verbal, and if verbal, how it was documented and witnessed). If your study included minors, state whether you obtained consent from parents or guardians."

3. Please include additional information regarding the survey or questionnaire used in the study and ensure that you have provided sufficient details that others could replicate the analyses. For instance, if you developed the survey or questionnaire as part of this study and it is not under a copyright more restrictive than CC-BY, please include a copy, in both the original language and English, as Supporting Information. If the questionnaire is published, please provide a citation to the (1) questionnaire and/or (2) original publication associated with the questionnaire.

4. Thank you for stating the following in the Competing Interests/Financial Disclosure * (delete as necessary) section:

“Authors CMW and KP have previously received research funding specific to children’s footwear research from Bobux International, administered through Monash University. KP has previously received research funding from ASICS Oceania specific to children’s and adolescent footwear research.

HB previously received in-kind support from Skobi Shoes (Foot Keepers Pty Ltd) specific to children’s footwear research administered through University of South Australia.

MH, SM, KG and EH have not received any support from any organisation for the submitted work, nor financial relationships with any organisations that might have an interest in the submitted work.

SB is employed by Bobux International.”

We note that you received funding from a commercial source: Bobux International & ASICS Oceania

Additional Editor Comments (if provided):

Dear Author

Your manuscripts need minor revisions. Please adhere to the feedback from the reviewers strictly.

Thank you.

Reviewers' comments:

Reviewer's Responses to Questions

**Comments to the Author**

1. Is the manuscript technically sound, and do the data support the conclusions?

Reviewer #1: Yes

Reviewer #2: Yes

2. Has the statistical analysis been performed appropriately and rigorously? 

Reviewer #1: Yes

Reviewer #2: Yes

3. Have the authors made all data underlying the findings in their manuscript fully available?

Reviewer #1: Yes

Reviewer #2: Yes

4. Is the manuscript presented in an intelligible fashion and written in standard English?

Reviewer #1: Yes

Reviewer #2: Yes

5. Review Comments to the Author

Reviewer #1: The study sounds original and useful, especially the method used (Delphi method) is interesting to be used for such studies. I would recommend adding a table gives some information about participants of the study (for example: number of participants, their profession, age, years of experience …etc.) as much information about participants that you have could be added, this will provide better understanding. Also, if possible adding few tables for each round to show the progress of the answers for each round.

Reviewer #2: It is a very interesting study. Please see my comments in the attachment file. The main recommendations are about the validity of questions on the first survey and the need of discussed the findings reported on S2 and S3 tables.

6. PLOS authors have the option to publish the peer review history of their article (what does this mean?). If published, this will include your full peer review and any attached files.

Reviewer #1: **Yes: **Nazdar Qudrat Abas

Reviewer #2: **Yes: **Angela Fernanda Espinosa

---

## [Author Response · Author response to Decision Letter 0]

15 Aug 2021

Reviewer comments

Reviewer #1:The study sounds original and useful, especially the method used (Delphi method) is interesting to be used for such studies. 

 Response: Thank you for your interest and kind words about the benefits of the study. 

I would recommend adding a table gives some information about participants of the study (for example: number of participants, their profession, age, years of experience …etc.) as much information about participants that you have could be added, this will provide better understanding. 

Response: We have provided further clarification at newly added Table 2. In addition, the sentence has been added to the manuscript with reference to this at Lines 317-320. 

Table 2 provides further details about the health professionals who participated, including a breakdown of the professions, average number of young children treated per month and years of experience. The number of participants in each round is shown in Fig 3.

Also, if possible adding few tables for each round to show the progress of the answers for each round.

Response: We have clarified Tables 3 and 4 within the manuscript. Readers are now oriented to highlight the number of participants retained through three rounds. The colour coding also applied in this table allows the reader to see the progress of each statement through each round based on the colour legend. To ensure the reader understands this progression, this additional information is provided within the body of the manuscript at Lines 347-350 for Round 1 and 374-376 for Rounds 2 and 3 statements. This information was: 

Tables 3 and 4 provide all statements generated by participants and the frequency (%) of participants who provided the same response. Statements highlighted in green were those that meet consensus (>70% of all participants providing the same response).

Tables 3 and 4 provide an outline of the statements progressing through Round 2 and Round 3 using the frequency (%) and same colour coding system as Round 1. 

Reviewer #2

Round 1: I consider that is necessary that authors explain how the questions of the initial survey was designed. Are the validated questions? Equally, it is necessary that be explained how was chosen the models of shoes whose were presented to participants

Response: Due to there being no validated tool, the survey questions were developed and face validity obtained through authorship team input and pilot testing. Details of how the survey was developed has been provided at Lines 155-169. This was: 

A purpose-built survey was developed by the authorship team due to the novelty of the questions of interest. Face validity was tested during development by collecting multiple photos of footwear types from those currently in online advertising in Australia, the United Kingdom and the United States. All authors then reviewed pictures of the types of footwear and agreed on grouping styles, that all grouping styles were represented, and the question phrasing for the target participants. The authorship group consisted of five clinician researchers, two parents with no research experience and one industry representative. All authors participated in all rounds of survey designs. Round 1 survey was then piloted with one parent and two health professionals and wording clarified based on their feedback.

Results: This apart is clear, only the S4Fig needs better definition

Response: We have checked the quality of the figure through the submission process to ensure it meets the journal quality assessment. 

Discussion: It would be very proper if the authors report the socioeconomic characteristics of participants that are parents. This variable is important in the moment to make a decision when is shopping shoes. One limitations that is important to discuss is that the variability of foot feature is wide and it is influenced by the phenotype and social and economic characteristics the parents and children. For example, in Latin America is common when a family has a lot of children and has no enough money that younger children use the shoes of the older siblings, which can influence in the feature of foot of the younger sons.

Response: Thank you for this response. We have considered the importance of this comment and regret we did not include factors within the construct of the survey, particularly socioeconomic characteristic including income, education, safety and social security factors. This means we are unable to comment further despite the importance of these factors. We have addressed this in the limitations and will consider these factors in data collection from parents in the future when conducting research on footwear choice. We have highlighted this at Lines 420-423

Lastly, we did not collect socioeconomic information from parent participants which may play a role in choices about footwear types and the opinions on how these choices impact the child. These factors could have been explored through collection of household income, education level of parent completing the survey and total number of children within the family. 

It is important that authors discus about the main findings reported on S2 and S3 tables

Response: We have included further clarity linking the information provided within tables previously named Table 2 and 3 (Now Tables 3 and 4) within the discussion and what this means in research, clinical care and for footwear designers. These points are at Line 413-417 and Lines 440 – 443.

Journal Requirements:

1. Please ensure that your manuscript meets PLOS ONE's style requirements, including those for file naming. The PLOS ONE style templates can be found at…

Response: We have checked and ensure all style requirements have been met to the best of our ability. 

2. Please provide additional details regarding participant consent. In the ethics statement in the Methods and online submission information, please ensure that you have specified what type of consent you obtained (for instance, written or verbal, and if verbal, how it was documented and witnessed). If your study included minors, state whether you obtained consent from parents or guardians."

Response: Further information is provided at Line 124 where participants provided written informed consent online. No minors were recruited as part of this study. 

3. Please include additional information regarding the survey or questionnaire used in the study and ensure that you have provided sufficient details that others could replicate the analyses. For instance, if you developed the survey or questionnaire as part of this study and it is not under a copyright more restrictive than CC-BY, please include a copy, in both the original language and English, as Supporting Information. If the questionnaire is published, please provide a citation to the (1) questionnaire and/or (2) original publication associated with the questionnaire.

Response: We have provided the full survey in S1 File. Round 1 Survey. Rounds two and three surveys are provided in S2 File. Round 2 Survey and S3 File. Round 3 Survey 

Response: We have reviewed the reference list and there is one addition of a publication about definitions of therapeutic footwear for children we found had been published in the last week added. No retractions have been made.

---

## [Decision Letter · Decision Letter 1]

18 May 2022

Young children’s footwear taxonomy: an international Delphi survey of parents, health and footwear professionals

PONE-D-21-06208R1

Dear Dr. Williams,

We’re pleased to inform you that your manuscript has been judged scientifically suitable for publication and will be formally accepted for publication once it meets all outstanding technical requirements.

Kind regards,

Vanessa Carels

Staff Editor

PLOS ONE

Additional Editor Comments (optional):

Reviewers' comments:

Reviewer's Responses to Questions

**Comments to the Author**

1. If the authors have adequately addressed your comments raised in a previous round of review and you feel that this manuscript is now acceptable for publication, you may indicate that here to bypass the “Comments to the Author” section, enter your conflict of interest statement in the “Confidential to Editor” section, and submit your "Accept" recommendation.

Reviewer #1: All comments have been addressed

Reviewer #2: All comments have been addressed

2. Is the manuscript technically sound, and do the data support the conclusions?

Reviewer #1: Yes

Reviewer #2: Yes

3. Has the statistical analysis been performed appropriately and rigorously? 

Reviewer #1: Yes

Reviewer #2: Yes

4. Have the authors made all data underlying the findings in their manuscript fully available?

Reviewer #1: Yes

Reviewer #2: Yes

5. Is the manuscript presented in an intelligible fashion and written in standard English?

Reviewer #1: Yes

Reviewer #2: Yes

6. Review Comments to the Author

Reviewer #1: I appreciate your detailed work on the comments, and providing all the necessary information that enriches your valuable work. I am glad that it is very clear and coordinated for readers as the study is important for providing a scientific results for designing footwears for our children.

Wish you all the best of luck.

Reviewer #2: I consider that you addressed in a proper way the comments and suggestions. For next publication you can explore the associations with your main outcomes and the socioeconomic conditions of the parents

7. PLOS authors have the option to publish the peer review history of their article (what does this mean?). If published, this will include your full peer review and any attached files.

Reviewer #1: **Yes: **Nazdar Qudrat Abas

Reviewer #2: **Yes: **Angela Fernanda Espinosa Aranzales

---

## [Editor Report · Acceptance letter]

1 Jun 2022

PONE-D-21-06208R1 

Young children’s footwear taxonomy: an international Delphi survey of parents, health and footwear industry professionals 

Dear Dr. Williams:

I'm pleased to inform you that your manuscript has been deemed suitable for publication in PLOS ONE. Congratulations! Your manuscript is now with our production department. 

Kind regards, 

on behalf of

Dr. Vanessa Carels 

Staff Editor

PLOS ONE